# Detection of R.1 lineage severe acute respiratory syndrome coronavirus 2 (SARS-CoV-2) with spike protein W152L/E484K/G769V mutations in Japan

**Yosuke Hirotsu**[1]*, **Masao Omata**[2,3]

**1** Genome Analysis Center, Yamanashi Central Hospital, Fujimi, Kofu, Yamanashi, Japan, **2** Department of Gastroenterology, Yamanashi Central Hospital, Fujimi, Kofu, Yamanashi, Japan, **3** The University of Tokyo, Hongo, Bunkyo-ku, Tokyo, Japan

* hirotsu-bdyu@ych.pref.yamanashi.jp

**Data Availability Statement:** All relevant data are within the manuscript and its Supporting Information files.

## Abstract

We aimed to investigate novel emerging severe acute respiratory syndrome coronavirus 2 (SARS-CoV-2) lineages in Japan that harbor variants in the spike protein receptor-binding domain (RBD). The total nucleic acid contents of samples from 159 patients with coronavirus disease 2019 (COVID-19) were subjected to whole genome sequencing. The SARS-CoV-2 genome sequences from these patients were examined for variants in spike protein RBD. In January 2021, three family members (one aged in their 40s and two aged under 10 years old) were found to be infected with SARS-CoV-2 harboring W152L/E484K/G769V mutations. These three patients were living in Japan and had no history of traveling abroad. After identifying these cases, we developed a TaqMan assay to screen for the above hallmark mutations and identified an additional 14 patients with the same mutations. The associated virus strain was classified into the GR clade (Global Initiative on Sharing Avian Influenza Data [GISAID]), 20B clade (Nextstrain), and R.1 lineage (Phylogenetic Assignment of Named Global Outbreak [PANGO] Lineages). As of April 22, 2021, R.1 lineage SARS-CoV-2 has been identified in 2,388 SARS-CoV-2 entries in the GISAID database, many of which were from Japan (38.2%; 913/2,388) and the United States (47.1%; 1,125/2,388). Compared with that in the United States, the percentage of SARS-CoV-2 isolates belonging to the R.1 lineage in Japan increased more rapidly over the period from October 24, 2020 to April 18, 2021. R.1 lineage SARS-CoV-2 has potential escape mutations in the spike protein RBD (E484K) and N-terminal domain (W152L); therefore, it will be necessary to continue to monitor the R.1 lineage as it spreads around the world.

## Author summary

A novel coronavirus, severe acute respiratory syndrome coronavirus 2 (SARS-CoV-2), emerged in December 2019 in Wuhan, China. SARS-CoV-2 had evolved and spread around the world, threatening human life. Several mutations, which alter the viral fitness,

**Funding:** This study was supported by a Grant-in-Aid for the Genome Research Project from Yamanashi Prefecture (to Y.H. and M.O.), Grant-in-Aid for Early-Career Scientists 18K16292 (to Y.H.) and Grant-in-Aid for Scientific Research (B) 20H03668 (to Y.H.) from the Japan Society for the Promotion of Science (JSPS) KAKENHI, a Research Grant for Young Scholars (to Y.H.) from Satoshi Omura Foundation, the YASUDA Medical Foundation (to Y.H.), the Uehara Memorial Foundation (to Y.H.), and Medical Research Grants from the Takeda Science Foundation (to Y.H.). The funders had no role in study design, data collection and analysis, decision to publish, or preparation of the manuscript.

**Competing interests:** The authors have declared that no competing interests exist.

virulence, and transmissibility, were identified in SARS-CoV-2. Here, we detected R.1 lineage SARS-CoV-2 harboring mutations in spike protein. The R.1 lineage have spread at the beginning of 2021 in Japan. This lineage has potential escape mutations in the spike protein receptor-binding domain (E484K) and N-terminal domain (W152L). We also developed a novel TaqMan assay targeting the hallmark mutations occurring in the spike proteins of R.1 lineage for screening. Our data indicate that emergent variants are dominated by the natural selection and need to be monitored by genomic epidemiological research.

## Introduction

Most mutations that occur during viral evolution are neutral and not to influence viral properties. However, some mutations are selectively propagated owing to their positive influence on viral fitness, virulence, and transmissibility [1,2]. Severe acute respiratory syndrome coronavirus 2 (SARS-CoV-2) "Variants of Concern" have emerged within the last few months, largely belonging to three major lineages, B.1.1.7, B.1.351, and P.1 [3–6]. These emerging lineages are all characterized by multiple mutations in the SARS-CoV-2 spike protein, raising concerns that they may escape monoclonal antibody therapy or vaccine-elicited antibodies. The B.1.1.7 lineage is estimated to have emerged in late September 2020 and has become the dominant strain of SARS-CoV-2 in the United Kingdom [3,7,8]. The B.1.351 lineage has become the dominant strain of SARS-CoV-2 in South Africa, where it was first detected in October 2020 [4]. The P.1 lineage was first identified in four travelers from Brazil and has been associated with cases of reinfection [5,6,9].

The hallmark mutation shared by the B.1.1.7, B.1.351, and P.1 lineages is N501Y, located in the receptor-binding domain (RBD) of the spike protein [3]. SARS-CoV-2 variants with this mutation are thought to be more transmissible and possibly more virulent [7,10–13]. The other hallmark mutation of the B.1.351 and P.1 lineages is E484K, which has been shown to reduce the neutralizing activity of antibodies, thus raising concerns about vaccine efficacy [14–19].

In this study, we conducted a genetic surveillance and identified the SARS-CoV-2 R.1 lineage, which harbors an E484K mutation in the RBD, via whole genome sequencing and TaqMan assay. We report the first detailed genetic characterization of the SARS-CoV-2 R.1 lineage, its familial transmission, and its viral evolution as assessed by phylogenetic analysis. The SARS-CoV-2 R.1 lineage has spread worldwide and remains especially prevalent in Japan.

## Materials and methods

### Ethics statement

The Institutional Review Board of the Clinical Research and Genome Research Committee at Yamanashi Central Hospital approved this study and the use of an opt-out consent method (Approval No. C2019-30). The requirement for written informed consent was waived owing to it being an observational study and the urgent need to collect COVID-19 data.

### Patients and sample collection

A total of 192 patients in our hospital were confirmed to have coronavirus disease 2019 (COVID-19), 159 of whom were selected to provide samples for subsequent genome analysis. Nasopharyngeal swab samples were collected by using cotton swabs and placed in 3 ml of viral

transport media (VTM) purchased from Copan Diagnostics (Murrieta, CA, United States). We used 200 µl of VTM for nucleic acid extraction, performed within 2 h of sample collection.

## Quantitative reverse transcription-polymerase chain reaction (RT-qPCR)

Total nucleic acid was isolated from the nasopharyngeal swab samples using the MagMAX Viral/Pathogen Nucleic Acid Isolation Kit (Thermo Fisher Scientific; Waltham, MA, United States) [20,21]. Following the protocol developed by the National Institute of Infectious Diseases in Japan [22], we performed one-step RT-qPCR to detect SARS-CoV-2 on a StepOnePlus Real-Time PCR System (Thermo Fisher Scientific). The viral load, measured as absolute copy number, was determined using a serially diluted DNA control targeting the *nucleocapsid* gene of SARS-CoV-2 (Integrated DNA Technologies; Coralville, IA, United States) [23,24].

## Whole-genome sequencing

SARS-CoV-2 genomic RNA was reverse transcribed into cDNA and amplified by using the Ion AmpliSeq SARS-CoV-2 Research Panel (Thermo Fisher Scientific) on the Ion Torrent Genexus System in accordance with the manufacturer's instructions [25]. Sequencing reads were processed, and their quality was assessed by using Genexus Software with SARS-CoV-2 plugins. The sequencing reads were mapped and aligned by using the torrent mapping alignment program. After initial mapping, a variant call was performed by using the Torrent Variant Caller. The COVID19AnnotateSnpEff plugin was used for the annotation of variants. Assembly was performed with the Iterative Refinement Meta-Assembler [26].

## Clade and lineage classification

The viral clade and lineage classifications were conducted by using the Global Initiative on Sharing Avian Influenza Data (GISAID) database [27], Nextstrain [28], and Phylogenetic Assignment of Named Global Outbreak (PANGO) Lineages [29]. The sequences of the three initially identified R.1 variants were deposited in the GISAID EpiCoV database (Accession Nos. EPI_ISL_1164927, EPI_ISL_1164928, and EPI_ISL_1164929).

Global sequencing data on the SARS-CoV-2 R.1 variant through April 22, 2021 were exported from the GISAID EpiCoV database. We searched for lineage "R.1" and found 2,388 available metadata entries for the R.1 lineage for the period from October 24, 2020 to April 18, 2021, when excluding two data entries lacking a specific collection date. We also searched for the total number of samples by applying the following parameters: host "human", location "Asia /Japan" or "North America/USA", collection date "October 24, 2020 to April 18, 2021". During this period, a total of 17,432 samples were registered in Japan, and 235,373 samples were registered in the United States.

## Rapid detection of R.1 lineage SARS-CoV-2 samples with a novel TaqMan assay

We designed a Custom TaqMan assay for detecting SARS-CoV-2 spike protein with the W152L and G769V mutations (Thermo Fisher Scientific). We also used a TaqMan SARS-CoV-2 Mutation Panel for detecting spike E484K (ID: ANU7GMZ, Thermo Fisher Scientific). TaqPath 1-Step RT-qPCR Master Mix CG was used as the master mix. The TaqMan MGB probes for the wild-type and variant alleles were labelled with VIC dye and FAM dye fluorescence, respectively.

## Results

### Household transmission of SARS-CoV-2 harboring a spike protein with the E484K mutation

As part of our ongoing genomic surveillance of SARS-CoV-2, we began to investigate SARS-CoV-2 spike protein RBD variants in Kofu city, Japan [25]. We previously identified P.1 lineage SARS-CoV-2, which harbors the K417T/E484K/N501Y mutations, in one patient [25]. A consecutive analysis detected the W152L/E484K/G769V mutations in three patients who provided samples on January 14, 2021. All three patients belonged to the same family: a man in his 40s (the father), a boy under 10 years old, and a girl under 10 years old. This family lived in Japan and had no history of travel to foreign countries. The mother of this family was also infected with SARS-CoV-2 around the same time as the other family members, but we were unable to obtain her samples because she was tested at another hospital. These results suggest that the family members were infected with the same virus lineage and that household transmission occurred.

### Genetic characterization defines the SARS-CoV-2 R.1 lineage

Our sequencing analysis of the SARS-CoV-2 isolates from the three family members identified the same 21 mutations in each isolate; these comprised 13 missense, six synonymous, and two intergenic variants. Among the missense mutations, four were in the spike protein (W152L, E484K, D614G, and G769V), four were in ORF1ab (T4692I, N6301S, L6337M, and I6525T), one was in the membrane protein (F28L), and four were in the nucleocapsid protein (S187L, R203K, G204R, and Q418H).

To examine the phylogeny based on the identified genetic variations, we analyzed SARS-CoV-2 genomic data using GISAID, Nextstrain, and Pangolin. The sequences of our SARS-CoV-2 were assigned as GR clade (GISAID), 20B clade (Nextstrain), and R.1 lineage (PANGO lineage). The R.1 lineage is a sublineage of the B.1.1.316 lineage, and the common mutations found in the R.1 lineage are listed in Table 1. For the rapid detection of SARS-CoV-2 R.1 lineage isolates, we designed a TaqMan assay that detects the hallmark spike protein mutations (W152L, E484K, and G769V) (Table 1). This TaqMan assay can successfully discriminate the wild-type and variant alleles in each position (Fig 1). Using the TaqMan assay, we identified an additional 14 patients who were infected with SARS-CoV-2 R.1 lineage. We further confirmed the results via whole genome sequencing. These findings suggest that the

**Table 1. Common mutations in the SARS-CoV-2 R.1 lineage.**

| Gene | Mutation |
| --- | --- |
| *ORF1b* | P314L |
| *ORF1b* | G1362R |
| *ORF1b* | P1936H |
| *S* | W152L |
| *S* | E484K |
| *S* | D614G |
| *S* | G769V |
| *N* | S187L |
| *N* | R203K |
| *N* | G204R |
| *N* | Q418H |

*ORF*, open reading frame; *S*, spike; *N*, nucleocapsid

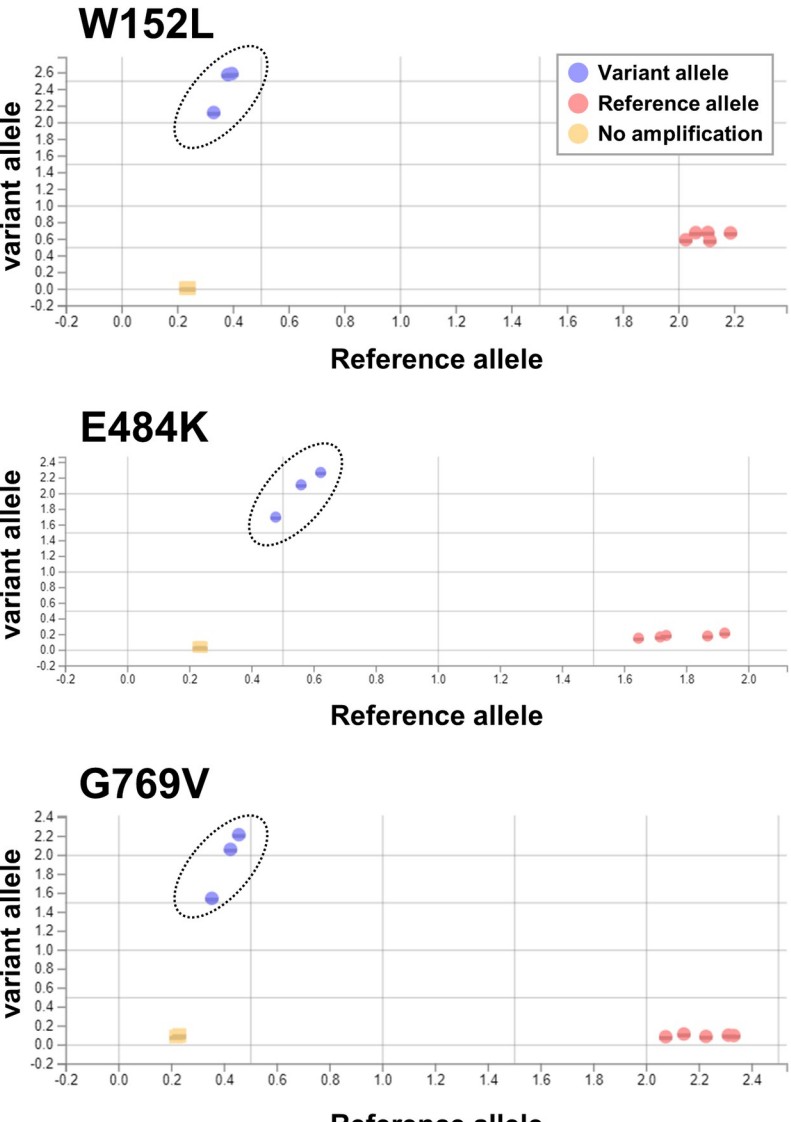

**Fig 1. TaqMan assay for discriminating R.1 lineage SARS-CoV-2 by its spike variant allele.** We analyzed samples of R.1 lineage SARS-CoV-2 (n = 3), which harbors a spike variant with the W152L/E484K/G769V mutations and samples of SARS-CoV-2 without these three mutations (n = 5) using a TaqMan assay. An allelic discrimination plot of the results, showing the spike variants with W152L (upper), E484K (middle), and G769V (lower). The dots indicate the mutant alleles (purple), reference allele (pink), or no amplification (orange). The dotted circles indicate the mutant alleles of each variant.

novel TaqMan assay targeting SARS-CoV-2 R.1 lineage hallmark mutations is a useful tool for detecting the presence of R.1 lineage isolates in SARS-CoV-2-positive PCR samples.

## Epidemiological event of SARS-CoV-2 R.1 lineage

To investigate the global distribution of SARS-CoV-2 R.1 lineage, we next collected registration data from the EpiCoV of GISAID database [27]. As of March 5, 2021, a total of 2,388 SARS-CoV-2 samples with R.1 lineage had been registered from all over the world, with the majority being from Japan (38.2%; 913/2,388) or the United States (47.1%; 1,125/2,388) (Fig 2A and Table 2). R.1 lineage SARS-CoV-2 was first reported at the end of October 2020 in

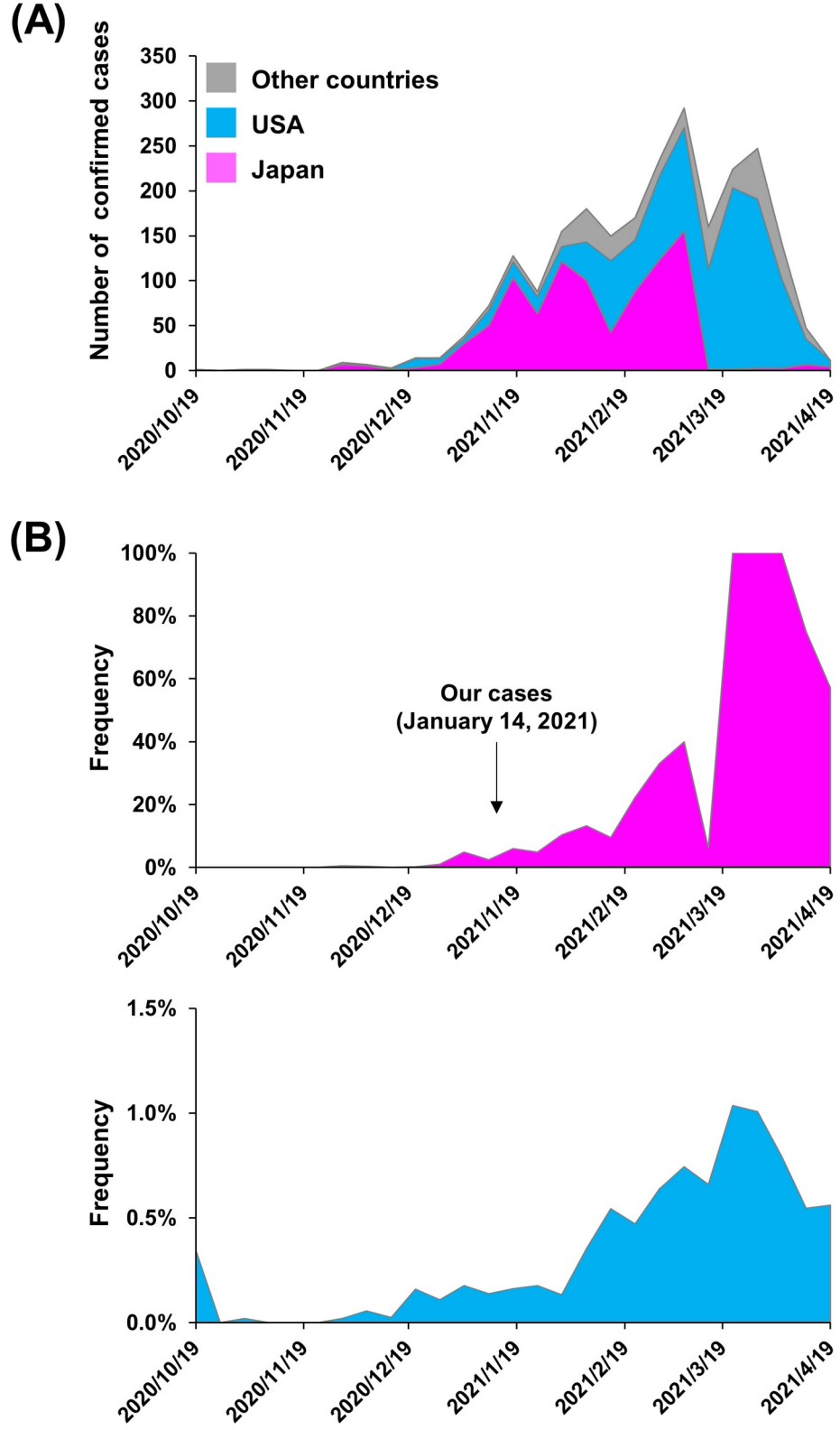

**Fig 2. Timeline of SARS-CoV-2 R.1 lineage emergence and country-specific percentages of global cases. (A)** Timeline of the number of confirmed cases of infection with R.1 lineage SARS-CoV-2. The plot, created based on the data in Table 1, shows the case load for Japan (pink), the United States (light blue), and other countries (gray). **(B)** The percentage of R.1 lineage SARS-CoV-2 strains relative to the total number of registered samples during the period is

shown for Japan (upper panel) and the United States (lower panel). The arrow indicates the date of infection for the initial three individuals infected with R.1 lineage SARS-CoV-2 who were identified in this study.

Texas, United States and was first detected in Japan at the end of November 2020. The change in the number of R.1 lineage SARS-CoV-2 samples has followed similar trends in the United States, Japan, and other countries (Fig 2A).

Regarding the period from October 24, 2020 to April 18, 2021, the number of R.1 lineage SARS-CoV-2 samples deposited in GISAID began an ongoing increase in February 2021 (Fig 2B), with the percentage of these isolates in Japan exceeding 20% since the beginning of March 2021 (Fig 2B). While the percentage of R.1 lineage SARS-CoV-2 remained low in the United States, it grew rapidly in Japan (Fig 2B). We also observed that the numbers and percentage of R.1 lineage SARS-CoV-2 isolates increased in our district of Japan (S1 Fig).

## Phylogenetic analysis of SARS-CoV-2 R.1 lineage

To determine the timing of the emergence of the SARS-CoV-2 R.1 lineage and its acquisition of characteristic mutations, we analyzed a phylogeny of SARS-CoV-2 carrying the E484K mutation, generated from global data. It revealed that the SARS-CoV-2 R.1 lineage forms a

**Table 2. Global numbers and percentages of confirmed R.1 lineage SARS-CoV-2 cases.**

| Country | Number of confirmed cases ($n$ = 2,388) | Frequency |
|---|---|---|
| USA | 1,125 | 47.1% |
| Japan | 913 | 38.2% |
| Germany | 100 | 4.2% |
| Austria | 69 | 2.9% |
| Belgium | 35 | 1.5% |
| Nether lands | 30 | 1.3% |
| Sweden | 26 | 1.1% |
| United Kingdom | 19 | 0.8% |
| Trinidad and Tobago | 18 | 0.8% |
| Spain | 12 | 0.5% |
| France | 8 | 0.3% |
| Canada | 7 | 0.3% |
| Ghana | 6 | 0.3% |
| Turkey | 3 | 0.1% |
| Australia | 3 | 0.1% |
| Croatia | 2 | 0.1% |
| Switzer land | 2 | 0.1% |
| Nigeria | 1 | 0.04% |
| China | 1 | 0.04% |
| United Arab Emirates | 1 | 0.04% |
| Denmark | 1 | 0.04% |
| Finland | 1 | 0.04% |
| Italy | 1 | 0.04% |
| Portugal | 1 | 0.04% |
| Slovakia | 1 | 0.04% |
| New Zealand | 1 | 0.04% |
| Aruba | 1 | 0.04% |
| Total | 2,388 | 100% |

monophyletic clade (Figs 3A and S2). The parental lineage harboring the E484K mutation later acquired spike protein W152L and G769V mutations, and the SARS-CoV-2 R.1 lineage was predicted to have emerged around September 9, 2020 (Fig 3A). Subsequently, a sublineage, which has been observed in Austria and the United States, diverged after acquiring the ORF1b G814C mutation around October 19, 2020 (Fig 3A). A parental lineage without the ORF1b G814C mutation has been prevalent mainly in Japan and the United States (Fig 3A). In addition to being detected in Japan, the SARS-CoV-2 R.1 lineage has been observed in 27 additional countries worldwide (Fig 3B and Table 2). Collectively, these results demonstrate that the SARS-CoV-2 R.1 lineage is spreading rapidly, especially in Japan.

## Discussion

COVID-19 vaccines have been approved in many countries within a year of the initial appearance of this disease, which is an unprecedented scientific achievement [30]. However, the emergence of mutations in SARS-CoV-2 spike RBD and N-terminal domain (NTD) are of great concern because of their potential contribution to immune escape [31,32]. In the present study, we observed an expansion in Japan of R.1 lineage SARS-CoV-2 harboring the spike RBD E484K and spike NTD W152L mutations, which have potential significance for immune escape. Previous reports showed that COVID-19 convalescent and mRNA vaccine-elicited sera/plasma have reduced neutralizing activity against SARS-CoV-2 harboring the E484K mutation [17–19]. Worryingly, the E484K mutation has been identified in SARS-CoV-2 "Variants of Concern", such as B.1.351 (South Africa) and P.1 (Brazil), and in SARS-CoV-2 "Variants of Interest", such as B.1.526 (New York, NY, USA), B.1.525 (United Kingdom/Nigeria), and P.2 (Brazil) [33–35]. Furthermore, the W152L mutation is located in the N3 loop of the NTD and may be related to decreased antibody neutralization activity [36]. The new emergent lineage B.1.429, which has a substitution in the same codon 152 (W152C), was first identified in California, United States; this lineage is defined as a "Variant of Interest" by the US Centers for Disease Control and Prevention (CDC) [35]. The W152C mutation is located in the NTD antigenic supersite (designated site i) and is also associated with reduced recognition of the NTD by neutralizing monoclonal antibodies [32,37].

There is concern that SARS-CoV-2 with these immune escape mutations may evade the host immune response. In Brazil, a patient was re-infected with SARS-CoV-2 carrying the E484K mutation, indicating that this mutation is related to escape from neutralizing antibodies in recovered patients [38]. Additionally, a patient who received the second shot of mRNA-1273 vaccine (Moderna) was infected with SARS-CoV-2 harboring the E484K mutation, despite having a serum neutralizing antibody titer that is normally sufficient to prevent infection [39]. Recently, R.1 lineage SARS-CoV-2 was detected at a skilled nursing facility in Kentucky, in residents and healthcare personnel who had received BNT162b2 mRNA vaccines (Pfizer-BioNTech) [40]. These results indicate that even after COVID-19 vaccination, the risk of infection is not completely eliminated. Although COVID-19 mRNA vaccines have demonstrated high efficacy for decreasing the risk of SARS-CoV-2 transmission and severe outcomes from COVID-19 [41–43], it is still necessary to monitor vaccinated individuals for new emergent lineages that have acquired novel escape mutations.

There are limitations to this epidemiological report on the SARS-CoV-2 R.1 lineage. The amount of SARS-CoV-2 sequencing data available from different countries does not reflect the actual prevalence of specific variants because the extent of sequencing analysis conducted varies among countries [30]. If the sequences of nearly all SARS-CoV-2-positive PCR specimens is analyzed, the results can represent the whole picture of prevailing virus lineages; unfortunately, this is not the case here. However, although the number of samples in our study is

**(A)**

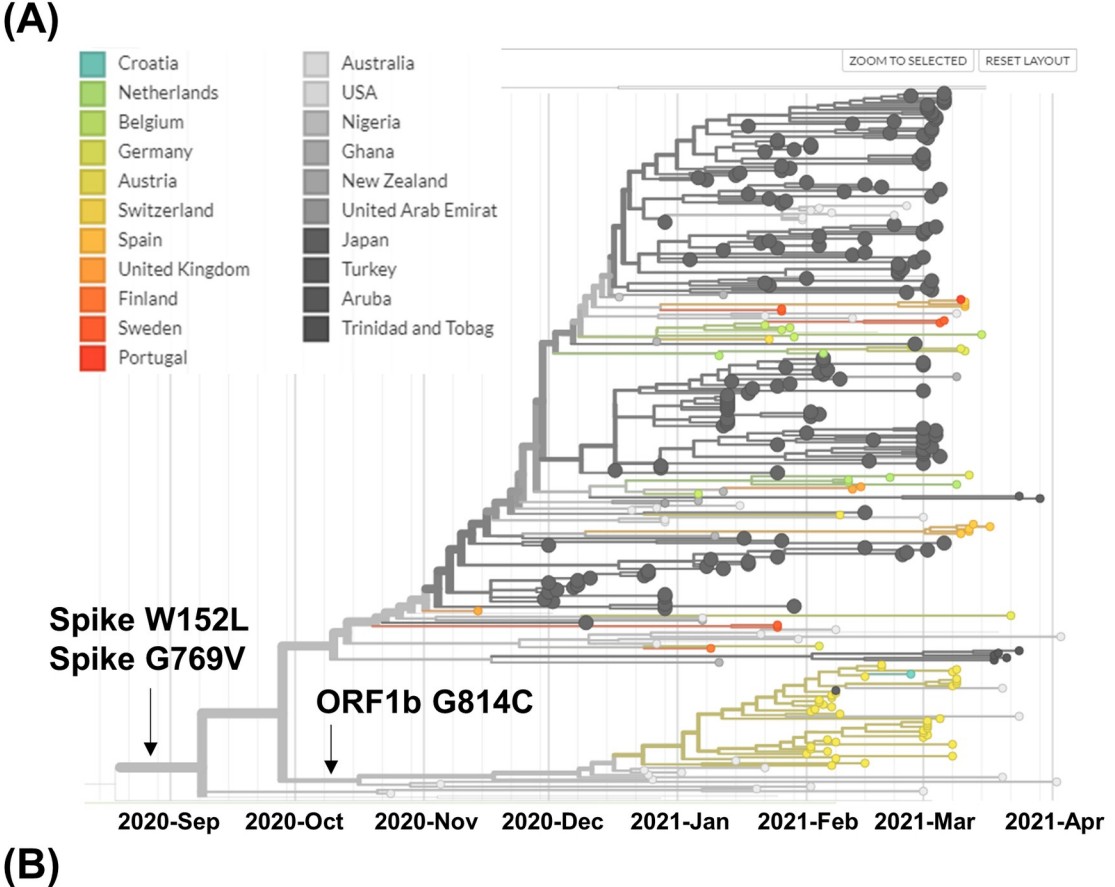

**(B)**

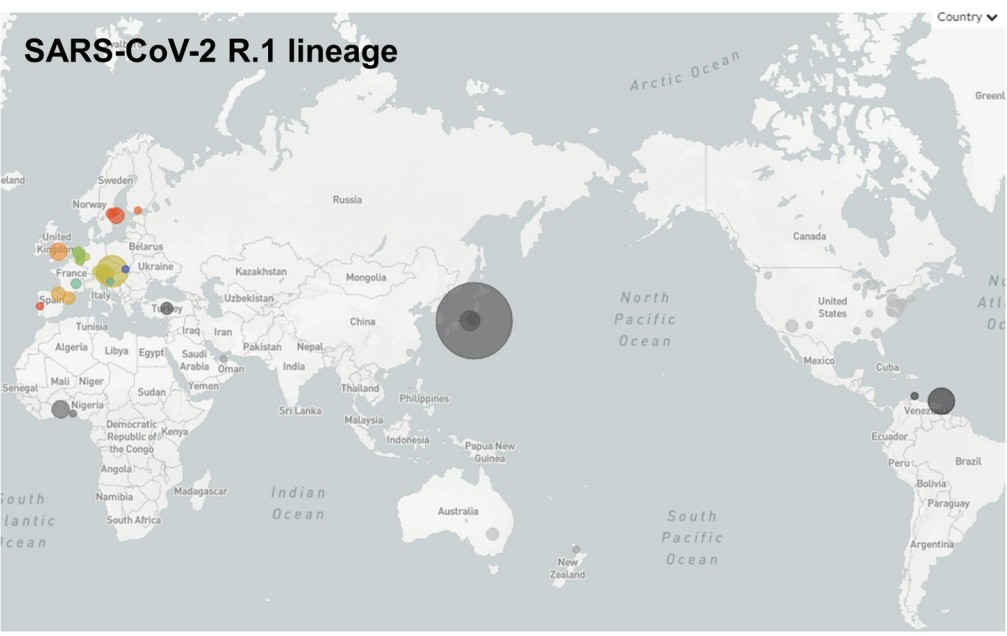

**Fig 3. Phylogenetic tree of R.1 lineage SARS-CoV-2. (A)** Global data on the SARS-CoV-2 R.1 lineage prevalence over time. The R.1 lineage acquired its spike W152L and G769V mutations at the root. The sublineage harbors an ORF1b G814C mutation. **(B)** Geographic distribution of the SARS-CoV-2 R.1 lineage. World map showing the geographic distribution of SARS-CoV-2 R.1 lineage as of April 22, 2021. The size of the circle indicates the number of samples. Nextstrain, https://nextstrain.org, CC-BY-4.0 license.

small, we analyzed all the SARS-CoV-2-positive PCR samples in our hospital and found that the percentage of samples with R.1 lineage SARS-CoV-2 increased to about 50% by the end of the study period (S1 Fig). Along with the R.1 lineage prevalence, the B.1.1.7 (United Kingdom) lineage prevalence is also increasing in Japan [44]. Further investigation is needed to determine whether these two lineages of SARS-CoV-2 will circulate dominantly in Japan.

Circulating SARS-CoV-2 acquires approximately one or two mutations per month. This seems to be very slow; however, the more the virus circulates in population, the more opportunity it has to change [45]. Therefore, genomic surveillance in real-time will provide us with important insights into the effectiveness of vaccines, the development of antibody therapy, and public health.

## Supporting information

**S1 Fig. Changes in the number of R.1 lineage SARS-CoV-2 strains identified in Kofu city, Yamanashi, Japan. (A)** The number of R.1 lineage SARS-CoV-2 samples identified by Taq-Man assay and whole genome analysis during the period from January to April 2021. The pink color indicates R.1 lineage strains, and the gray color indicates other strains. **(B)** The percentage of SARS-CoV-2 strains detected over the period from January to April 2021 that belong to the R.1 lineage.
(TIF)

**S2 Fig. Global data of SARS-CoV-2 harboring spike protein with an E484K mutation as of April 22, 2021.** The dotted pink line shows a monophyletic clade containing the SARS-CoV-2 R.1 lineage.
(TIF)

**S1 File. Acknowledgments to the authors and laboratory for registering the genome data via GISAID.**
(PDF)

## Acknowledgments

We also thank Masato Kondo, Ryota Tanaka, Kazuo Sakai, Manami Nagano, Takuhito Fukami, and Ryo Kitamura (Thermo Fisher Scientific) for technical help, all of the medical and ancillary hospital staff for their support, and the patients for their participation. We thank all researchers who share genome data on GISAID (http://www.gisaid.org) (S1 File). We thank Katie Oakley, PhD, from Edanz Group (https://jp.edanz.com/ac) for editing a draft of this manuscript.

## Author Contributions

**Conceptualization:** Yosuke Hirotsu, Masao Omata.

**Data curation:** Yosuke Hirotsu.

**Formal analysis:** Yosuke Hirotsu.

**Funding acquisition:** Yosuke Hirotsu.

**Investigation:** Yosuke Hirotsu.

**Project administration:** Yosuke Hirotsu.

**Resources:** Yosuke Hirotsu.

**Supervision:** Masao Omata.

**Writing – original draft:** Yosuke Hirotsu.

**Writing – review & editing:** Yosuke Hirotsu, Masao Omata.

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
