## [Decision Letter · Decision Letter 0]

19 Apr 2021

Dear Dr. Hirotsu,

Thank you very much for submitting your manuscript "Household transmission of SARS-CoV-2 R.1 lineage with spike E484K mutation in Japan" for consideration at PLOS Pathogens. As with all papers reviewed by the journal, your manuscript was reviewed by members of the editorial board and by several independent reviewers. The reviewers appreciated the attention to an important topic. Although this kind of observational study is not the traditional research type article that PLoS Pathog publishes, we are considering a short/discovery report type format for observations that are of particular significance and/or broad impact.  

We believe the authors' observation falls into that category. Based on the reviews, we are likely to accept this manuscript for publication, providing that you modify the manuscript according to the review recommendations.

Please address the comments of both reviewers constructively.  We concur that addition of a well-constructed  phylogenetic tree to show that the R1 lineage forms a monophyletic clade would be important (Reviewer #2 comments).  This editor also suggests that the authors discuss the potential significance of the W152L mutation. Note that W152C is a defining mutation in B.1.429, an emerging Variant of Interest, and is located in the NTD antigenic supersite i (PMID: 33761326, 33821281).  In addition, please ensure that the manuscript is carefully proof-read to correct for grammar, spelling and syntax. Some incomplete sentences (e.g line 152-154) detract from a major point that the authors might want to make. With regards to the Japanese sequence data in Fig. 2B, please update data to Mar 1 if available.   

Sincerely,

Benhur Lee

Section Editor

PLOS Pathogens

Benhur Lee

Section Editor

PLOS Pathogens

Kasturi Haldar

Editor-in-Chief

PLOS Pathogens

orcid.org/0000-0001-5065-158X

Michael Malim

Editor-in-Chief

PLOS Pathogens

orcid.org/0000-0002-7699-2064

Reviewer Comments (if any, and for reference):

Reviewer's Responses to Questions

**Part I - Summary**

Reviewer #1: This is a straightforward case study describing household spread in Japan of the R.1 lineage of SARS-CoV-2. The study is generally well written. It is quite medically oriented and rather observational. There are no experiments.

Reviewer #2: This paper gives an initial description of a novel SARS-CoV-2 lineages, R.1, which appears to be mainly circulating in the USA and Japan, with a focus on the cases in Japan. R.1 carries the E484K mutation in spike, also shared by the 501Y.V2 and 501Y.V3 variants of concern, first identified in South Africa & Brazil, respectively, which is thought to be associated with possibly evading immune response, allowing reinfection and/or vaccine evasion.

This is a clear and well-written paper, presented in a level-headed manner. It is a descriptive work of a novel lineage, so there is no great detail about changes in viral behaviour or speculation of origins, but for a short initial description I do not think that is necessarily to be expected. The methods and results are detailed well and sufficient information is given to identify the lineage clearly in other sequences and to gain an understanding of the lineage and its spread so far.

I am not so familiar with the new format I believe this was submitted under therefore will leave to the editor to determine if this satisfied those requirements, but I believe this is a scientifically sound paper giving a good description of a novel SARS-CoV-2 lineage. I would recommend only 1 major and 1 minor changes before acceptance.

**Part II – Major Issues: Key Experiments Required for Acceptance**

Reviewer #1: Epidemiological event of R.1 lineage section should discuss sampling bias.

Reviewer #2: I believe this paper is a good description of the R.1 lineage, and useful to those who wish to identify it and understand its initial distribution and spread. However, I think this paper would benefit greatly from even a very basic phylogenetic tree of the lineage and the sequences it covers. I think this would allow even a quick look at the relationships between the global sequences, and to show that this is indeed one monophyletic clade.

From looking at this clade on the focal Nextstrain 484K build (https://nextstrain.org/groups/neherlab/ncov/S.E484?c=country&f_pango_lineage=R.1&label=mlabel:20B/C18877T), it does indeed seem to be, but sometimes in the paper it seems like perhaps this is presented as ambiguous (though this may not have been intentional, and I may be misinterpreting the writing - apologies if so). For example in lines 147-148, saying "...implying that [the lineage] may have emerged in several regions at approximately the same time."

Further a phylogeny would help to clarify the meaning of statement like in lines 168-169: "The circulating trend of R.1 lineage showed similar pattern (note: typo, should be 'patterns') in each country, implying that the ancestral strain acquired homegrown variants, and subsequently R.1 lineage are likely to emerge in several countries." Is there evidence of unique R.1 variants in different countries? (I actually think from the Nextstrain tree, there may be - but showing a tree would display this more clearly, if so.)

I think even a fairly basic phylogeny as an additional figure, with a small additional description of what can be seen, would add a lot of value to the paper in showing any clusters in any countries, the relationship between USA & Japanese sequences, and the genetic diversity of the cluster (including any sub-clusters within the lineage with S or other AA mutations).

**Part III – Minor Issues: Editorial and Data Presentation Modifications**

Reviewer #1: There are a number of minor typos / grammar problems.

I also did not see a supplemental data table to conform with the GISAID terms of use.

Reviewer #2: I understand the world of SARS-CoV-2 research moves incredibly quickly, but I fear by the time this work is published it may seem quite outdated, given the data was collected through Feb 2021. As the work is descriptive I do think much of the value lies in the descriptions being as up-to-date as possible, so I would urge the authors to update the paper (counts, global coverage, etc) to cover data through as recent a period as is possible. This would also possibly shed some clarity onto statements like that of 153-154 ("it is possible that R.1 lineage is gradually increasing."

Minor typos/wording: Line 170 "acquired" should be "acquires". Line 174, should be "therefore, new emerging strain[s] [possibly] have been missed."

PLOS authors have the option to publish the peer review history of their article (what does this mean?). If published, this will include your full peer review and any attached files.

Reviewer #1: No

Reviewer #2: No

Figure Files:

Data Requirements:

Reproducibility:

References:

---

## [Editor Report · Decision Letter 1]

5 May 2021

Dear Dr. Hirotsu,

We are pleased to inform you that your manuscript 'Detection of R.1 lineage severe acute respiratory syndrome coronavirus 2 (SARS-CoV-2) with spike protein W152L/E484K/G769V mutations in Japan' has been provisionally accepted for publication in PLOS Pathogens.

In addition, we would like to encourage the authors to deposit their whole genome sequences of R.1 lineage viruses into properly curated databases like GISAID so that your efforts can be properly acknowledged. We believe your findings should add to the usefulness of such databases.  

Best regards,

Benhur Lee

Section Editor

PLOS Pathogens

Benhur Lee

Section Editor

PLOS Pathogens

Kasturi Haldar

Editor-in-Chief

PLOS Pathogens

orcid.org/0000-0001-5065-158X

Michael Malim

Editor-in-Chief

PLOS Pathogens

orcid.org/0000-0002-7699-2064

Reviewer Comments (if any, and for reference): The addition of the phylogenetic tree was edifying, in addition to the updated data showing the increasing spread of the R1 lineage in Japan. We encourage the authors to deposit  their full-length genomic sequence into properly curated databases like GISAID so that their efforts can be acknowledged.

---

## [Editor Report · Acceptance letter]

3 Jun 2021

Dear Dr. Hirotsu,

We are delighted to inform you that your manuscript, "Detection of R.1 lineage severe acute respiratory syndrome coronavirus 2 (SARS-CoV-2) with spike protein W152L/E484K/G769V mutations in Japan," has been formally accepted for publication in PLOS Pathogens.

Best regards,

Kasturi Haldar

Editor-in-Chief

PLOS Pathogens

orcid.org/0000-0001-5065-158X

Michael Malim

Editor-in-Chief

PLOS Pathogens

orcid.org/0000-0002-7699-2064